



# To Bucket or not to Bucket? Analyzing the performance and interpretability of hybrid hydrological models with dynamic parameterization

Eduardo Acuña Espinoza [1], Ralf Loritz [1], Manuel Álvarez Chaves [2], Nicole Bäuerle [3], and Uwe Ehret [1]

[1]Institute of Water Resources and River Basin Management, Karlsruhe Institute of Technology (KIT), Karlsruhe, Germany
[2]Stuttgart Center for Simulation Science, Statistical Model-Data Integration, University of Stuttgart, Stuttgart, Germany
[3]Institute of Stochastics, Karlsruhe Institute of Technology (KIT), Karlsruhe, Germany

**Correspondence:** Eduardo Acuña Espinoza (eduardo.espinoza@kit.edu)

**Abstract.** Hydrological hybrid models have been proposed as an option to combine the enhanced performance of deep learning methods with the interpretability of process-based models. Among the various hybrid methods available, the dynamic parameterization of conceptual models using LSTM networks has shown high potential. We explored this method further to evaluate specifically if the flexibility given by the dynamic parameterization overwrites the physical interpretability of the process-based part. We conducted our study using a subset of CAMELS-GB dataset. First, we show that the hybrid model can reach state-of-the-art performance, fully comparable with a regional LSTM, and surpassing the performance of conceptual models in the same area. We then modified the conceptual model structure to assess if the dynamic parameterization can compensate for structural deficiencies of the model. Our results demonstrated the ability of the deep learning method to effectively compensate for deficiencies and implausible model structures in the hydrological models. This indicates that the hydrological model did not give a strong enough regularization to drop the hybrid model's performance. A model selection based purely on the performance to predict streamflow, for this type of hybrid model, is hence not advisable. However, this does not entail that such hybrid models cannot be used to gain a better understanding of a hydrological system by studying other hydrological fluxes and states than discharge. Comparisons with external data, as well as the internal functioning of the hybrid model, reiterate that if a well-tested model architecture is combined with a LSTM, the deep learning model can learn to operate the process-based model in a consistent matter. In conclusion, this study demonstrated that hybrid models, if set up cautiously, can combine the enhanced performance of deep learning methods while maintaining good interpretability in the process-based part.

## 1 Introduction

Rainfall-runoff models are useful tools to support decision-making processes related to water resources management and flood protection. Over the past decades, hydrological conceptual models have emerged as important tools for these purposes, finding widespread usage in academia, industry and national weather services. These models, known for their simplicity, computational efficiency, and ability to generalize, encode our understanding of hydrological processes within a fixed model structure. By connecting the various macroscopic storages (also known as buckets) through a network of fluxes, conceptual models try to





emulate the internal processes occurring within a catchment. The accurate representation of these processes relies on calibrated parameters, which are adjusted to achieve consistency with observed data. Examples of widely used conceptual models include Hydrologiska Byrans Vattenavdelning (HBV) (Bergström, 1976), Sacramento (Burnash et al., 1995), GR4J(Perrin et al., 2003), Precipitation-Runoff Modelling System (PRMS) (Leavesley et al., 1983) and TOPMODEL (Beven and Kirkby, 1979), to name a few. Additionally, there are software tools available, such as Raven (Craig et al., 2020) and Superflex (Dal Molin et al., 2021), which facilitate the creation of customized models tailored to specific basin characteristics and key hydrological processes.

Despite the widespread use of conceptual models, data-driven techniques, particularly long short-term memory (LSTM) (Hochreiter and Schmidhuber, 1997) networks, have recently shown the potential to outperform conceptual models, particularly in large sample model comparison studies (Kratzert et al., 2019b; Lees et al., 2021; Feng et al., 2020). The improvement in performance can be attributed, partly, to the inherent flexibility of LSTM networks, which surpasses the constraints imposed by fixed model structures by effectively mapping connections and patterns through optimization techniques. However, the characteristic that allows LSTMs to excel in performance has also sparked criticism regarding their interpretability (Reichstein et al., 2019), owing to the fact that states, weights and biasses in LSTMs lack clear semantic meaning, making it challenging to discern the underlying reasons for their decision and predictions. Despite notable advancements in linking hydrological concepts to the internal states of LSTMs (Kratzert et al., 2019a; Lees et al., 2022), concerns within the community towards their interpretability persist.

Reichstein et al. (2019) and Shen et al. (2023) indicate that combining process-based environmental models with ML approaches, into so-called hybrid models, can harness the strengths of both methodologies, leveraging the improved performance of data-driven techniques while retaining the interpretability and consistency offered by physical models. Among the various approaches proposed by the authors, one method involves the parameterization of physical models using data-driven techniques. Kraft et al. (2022) applied this method, with the idea that replacing poorly understood or challenging-to-parameterize processes with machine learning (ML) models can effectively reduce model biases and enhance local adaptivity. Moreover, their study demonstrated that the hybrid approach achieved comparable performance to process-based models. Feng et al. (2022) followed a similar procedure, in which the parameters of an HBV model were dynamically estimated using a LSTM network. Their study convincingly demonstrates the effectiveness of this approach, revealing its ability to achieve state-of-the-art performance that directly rivals purely data-driven methods when applied to the Catchment Attributes and Meteorology for Large-sample Studies (CAMELS) data set (Addor et al., 2017) in the United States (CAMELS-US). In their study, Feng et al. (2022) implemented both static and dynamic parameterization techniques and observed that the latter led to slightly improved performance. However, it is important to note that the authors warn about the flexibility of LSTM networks when used for dynamic parameterizations. They posed the hypothesis that, while applying dynamic parameterization increases the likelihood of achieving high performance, there is a risk of compromising the physical significance of the model, potentially resulting in the system behaving more like an LSTM variant rather than a hydrologically-meaningful model. In other words, model deficits and ill-defined process descriptions might be compensated by the LSTM.

Motivated by the outcomes achieved in the aforementioned articles, our study aims to dig deeper into the coupling of LSTM and conceptual models. We believe that dynamic parameters provided by an LSTM allow the conceptual model to not only





adapt to changes in the hydrological regime, which is physically reasonable (Loritz et al., 2018) but also to compensate for inherent deficiencies or oversimplifications within the model structure. More specifically, and guided by the warning given by Feng et al. (2022), our study aims to address the following research questions:

- 1. Do conceptual models serve as an effective regularization mechanism for the dynamic parameterization of LSTMs?

- 2. Does the data-driven dynamic parameterization compromise the physical interpretability of the conceptual model?

To address the research questions at hand, we have structured our article as follows: In Section 2 we describe the structure and training process of the conceptual, data-driven, and hybrid models employed in this study. Additionally, we outline the details of the dataset used to train and test the rainfall-runoff models. In Section 3, after proving that the hybrid model performance is comparable with the LSTM, we conduct experiments to answer our first research question. By systematically modifying the conceptual model, we assess how different forms of regularization affect the overall performance of the hybrid model. This will allow us to better understand the effect of different conceptual models as regularization and the interaction between the data-driven and conceptual components. Furthermore, to address the second research question, we analyze the internal states of the conceptual model to evaluate how much physical interpretability the different variants of our conceptual model is keeping. Finally, we summarize our key findings in Section 4.

## 2 Data and Methods

The first subsections of this unit present an overview of the dataset utilized for training and testing our models. The second subsection describes the dataset used to evaluate the internal states of our conceptual models. The last three segments explain the structure and training process of the conceptual, data-driven, and hybrid models, respectively.

### 2.1 CAMELS-GB

To train and test our rainfall-runoff models we used the CAMELS-GB dataset (Coxon et al., 2020). This dataset contains information about river discharge, catchment attributes, and meteorological time series for 671 catchments in Great Britain. We applied multiple filtering criteria to obtain the final subset of catchments used in this paper.

The first filtering criterion considered was data completeness, taking into account both meteorological and discharge information. While the dataset initially consisted of 671 catchments with meteorological data spanning from 1970 to 2015, not all basins had continuous discharge time series during this period. Therefore, a trade-off between the number of basins selected and data completeness was made. After analyzing the data, we chose a subset of 387 basins that had (almost) continuous discharge series from 1987 onward. This subset was used to train the reference LSTM as a regional model, following the recommendation proposed by Kratzert et al. (2019b). To train the conceptual and hybrid models, and to make a fair performance comparison between the three models, two additional filtering steps were applied.

The second filtering criterion considered an inherent limitation of conceptual models, which in principle does not affect data-driven techniques. Unlike LSTM networks, which learn directly from the data without a predefined structure, conceptual



models have fixed model architectures designed to represent specific processes. This means that anthropogenic impacts such
as reservoir operations, withdrawals, or transfers may not be adequately captured by conceptual models unless they are di-
rectly accounted for. While this limitation is a clear advantage of data-driven techniques, we aimed to ensure a fair comparison
between the models on a level playing field. Therefore, we selected only basins within the initial subset with the label "bench-
mark_catch=TRUE" which, according to Coxon et al. (2020), can be treated as "near-natural". In other words, catchments in
which the human influence in flow regimes is modest and where natural processes predominantly drive the flow regimes.

The last filtering criterion considered the temporal resolution of the data and the size of the catchment. The CAMELS-GB
dataset contains data with a daily resolution. Consequently, we need to consider catchments with a sufficient size such that
discharge variations are resolved by daily data. After applying the aforementioned filters, we identified 60 basins that passed
all three filters. Figure 1a shows the locations of these basins in the UK. For detailed information on the specific basin IDs and
corresponding time series data, please refer to the supplemental information provided with this article. As indicated before,
these catchments were used to train the conceptual and hybrid models, and to compare the performance among the three
modeling approaches.

Lastly, we would like to highlight that even though our filtering criteria reduced the size of our original dataset, we still
used 60 basins and 25 years of data per basin to conduct our study, which we suggest is sufficiently large to draw robust
conclusions from our study results. Furthermore, the spatial distribution of the selected catchments covers most of the original
range. Moreover, our models' performance, as demonstrated later, aligns with the benchmark set by Coxon et al. (2020) for
the CAMELS-GB full dataset, further validating the robustness of our methods. Lastly, as mentioned in the introduction and
elaborated in subsequent sections, besides the data-driven and hybrid models, we also trained the conceptual models. Therefore,
using a subset of the original dataset played an important role to maintain a moderate computational cost.

In accordance with ML practices, the data was divided into training, validation and testing sets. The training period, used
to calibrate the models, spans from 01/10/1987 to 30/09/1999 (12 years). The validation period, used to track the loss during
the calibration, used data from 01/10/1999 to 30/09/2004 (5 years). Lastly, the testing period, used to estimate the model
performance and carry out the evaluations, covered the period from 01/10/2006 to 30/09/2012 (6 years).

## 2.2  ERA5-Land

As outlined in the introduction, one of the primary objectives of this study is to assess the physical consistency of our hybrid
model. To achieve this, we conducted several tests, one of which involved comparing the unsaturated zone reservoir of the
conceptual model with soil moisture estimates (details in section 3.3). Following the procedure proposed by Lees et al. (2022)),
we compared our model's results with data from ERA5-LAND (Muñoz Sabater et al., 2021). This dataset, based on a 9 km x
9 km gridded format (see Figure 1b), is a land component reanalysis of the ERA5 dataset (Hersbach et al., 2020). According
to Lees et al. (2022), reanalysis data offers several advantages, including longer time series availability, easy transferability to
basin-average quantities (consistent with the CAMELS-GB process) due to the gridded format, and global coverage, enabling
its application in various locations. As our study region aligns with that of Lees et al. (2022) in Great Britain, we utilized
the NetCDF file provided by the authors, which was publicly accessible. However, to suit our specific analysis, we further





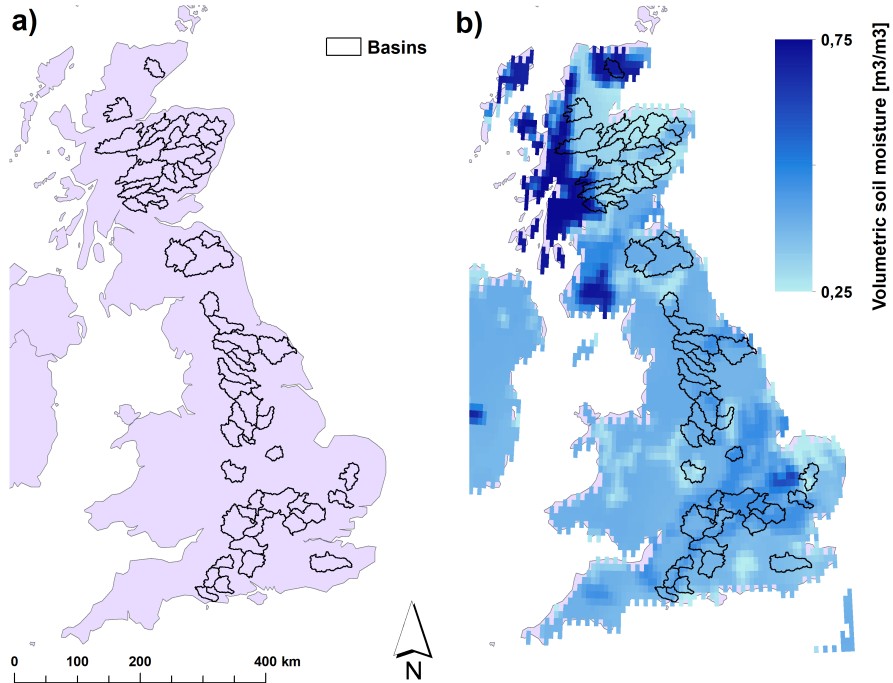

**Figure 1.** a) Location of the 60 basins selected from CAMELS-GB to carry out the experiments b) Example of ERA5-Land Soil Water Volume information

processed the data to obtain catchment-average estimates for our 60 basins of interest. Subsequently, we extracted values corresponding to our testing period and normalized the data to a [0-1] range for comparative purposes. This normalization

approach is consistent with Ehret et al. (2020), where they followed a similar process to assess the realism of the unsaturated zone soil moisture dynamics of a conceptual hydrological model.

ERA5-Land contains information about soil water volume at four different levels. Level 1 (swvl1) provides information at a depth of 0 cm to 7 cm, level 2 (swvl2) from 7 cm to 28 cm, level 3 (swvl3) from 28 cm to 100 cm, and level 4 (swvl4) from 100 cm to 289 cm. When using this information to evaluate our models, we consistently found higher correlations for all cases

when compared against swvl3. Therefore, the results reported in Section 3.3 are associated with that depth.

## 2.3  Conceptual hydrological model

In this study, we employ a conceptual model named "Simple Hydrological model" (SHM) (Ehret et al., 2020) that is in its essence a slightly altered HBV model. We will use SHM both as a stand-alone benchmark and as an integral component of the hybrid model. A detailed explanation of the model is available in Ehret et al. (2020). Nonetheless, we offer a concise

overview of the model's key features. In a slight variation from Ehret et al. (2020) we included a snow module, and the potential evapotranspiration is read directly from the CAMELS-GB dataset.





Figure 2 illustrates the overall structure of the model. The model input consists of three forcing variables: Precipitation (**P**) [$mm \cdot d^{-1}$], Temperature (**T**) [$°C$], and Potential Evapotranspiration (**ETp**) [$mm \cdot d^{-1}$]. Then, to emulate the hydrological processes occurring in the basin, the model uses five storage components, namely: snow module, unsaturated zone, fast flow, interflow, and baseflow. Overall, to regulate the fluxes between components, eight parameters need to be calibrated: *dd*, *f_thr*, *su_max*, *β*, *perc*, *kf*, *ki* and *kb* (see units in Table 2).

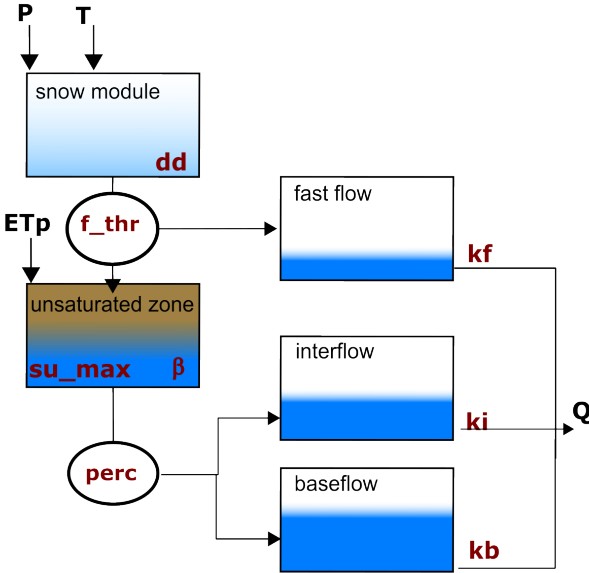

**Figure 2.** Structure of SHM hydrological model used for rainfall-runoff prediction

The snow module receives (**P**) and (**T**) as inputs. Based on the temperature, precipitation is either stored as snow or moves forward together with additional discharge from snowmelt (if any). Snowmelt is calculated using the degree-day method in which the parameter *dd* relates to the volume of snowmelt at a given temperature. If the outflow of the snow module exceeds a threshold (*f_thr*), the excess is directed to the fast flow reservoir while the remaining portion enters the unsaturated zone bucket. On the other hand, if the snow storage outflow is smaller than *f_thr*, all water enters the unsaturated zone as input. Within the unsaturated zone, several processes occur. First, evapotranspiration causes water loss. The potential evapotranspiration (**ETp**) is provided as a forcing variable but is adjusted to reflect the actual evapotranspiration considering water availability. Additionally, there is an outflow from the unsaturated zone, determined by a power relationship involving the parameters *su_max* and *β*. This outflow is then divided by the *perc* parameter, allocating portions to the inflows of the interflow and baseflow storages. Finally, the total discharge of the basin is computed as the sum of the outflows from the fast flow, interflow, and baseflow storages. Each outflow is a linear function of its corresponding storage and the recession parameters *kf*, *ki*, and *kb*, respectively.

To establish a benchmark for comparing our data-driven and hybrid methods, we performed individual calibrations of the SHM for each specific basin of interest (e.g. 60 SHM models). This approach is in line with Kratzert et al. (2019b) and Nearing



et al. (2021), who indicate that conceptual models generally perform better when calibrated at the individual basin level rather than using a regional calibration approach. To ensure a fair comparison and mitigate potential calibration biases that may favour our hybrid model, we employed three established calibration methods and selected the one that yielded the best performance for each basin. The first two calibration methods were Shuffled Complex Evolution (SCE-UA) (Duan et al., 1994) and the Differential Evolution Adaptive Metropolis (DREAM) (Vrugt, 2016), both implemented within the Spotpy library (Houska

et al., 2015). In addition, we calibrated the models using ADAM (Kingma and Ba, 2014), using the Automatic Differentiation subroutine in PyTorch (Paszke et al., 2017) to calculate the gradients.

## 2.4 LSTM

As mentioned in previous sections, we incorporated an LSTM model as a benchmark for our comparison. For a comprehensive understanding of the internal workings of LSTM networks, we refer to the work by Kratzert et al. (2018). In this subsection,

we will provide an overview of the key aspects required to comprehend the training process. Our data-driven model was implemented using the PyTorch library (Paszke et al., 2019), and the corresponding code can be found in the repository accompanying this paper.

The majority of hyperparameters of our model align with the findings of Lees et al. (2021). Our model uses a single LSTM layer with 64 hidden states, a dropout rate of 0.4, and an initial learning rate of 1e-3. We use a batch size of 256 and set the

initial bias of the forget gate to 3. Additionally, the ADAM algorithm was used for the optimization. In a slight deviation from Lees et al. (2021), we used a sequence length of 180 days, which aided in accelerating the training process.

The model input consisted of three dynamic forcing variables, which varied at each time step, along with 20 static attributes encoding key characteristics of the catchments (see Table 1 for details). The model output was compared against the observed specific discharge. In accordance with common ML practices, both the input and output data were standardized using the global

mean and standard deviation of the training dataset.

To optimize the LSTM parameters we used the basin-averaged Nash-Sutcliffe Efficiency ($NSE^*$) loss function proposed by Kratzert et al. (2019b). This function divides the squared error between the modeled and observed output by the variance of the specific discharge series associated with each respective basin in the training period. As described by the authors, $NSE^*$ provides an objective function that reduces bias towards large humid basins during the optimization process, avoiding the

underperformance of the regional model in catchments with lower discharges. Given that we are training our regional model for a batch size $N = 256$, the training loss was calculated according to equation 1:

$$NSE^* = \frac{1}{N} \cdot \sum_{i=1}^{N} \frac{(y_i^{obs} - y_i^{sim})^2}{(s_i + \epsilon)^2}, \tag{1}$$

where $y_i^{obs}$ is the observed discharged (standardized), $y_i^{sim}$ the simulated discharged (standardized), $s_i$ the standard deviation of the flow series (in training period) for the basin associated with element $i$, and $\epsilon$ is a numerical stabilizer ($\epsilon = 0.1$) so the

loss function remains stable even when basins with low flow standard deviations are considered.





**Table 1.** Dynamic variables and static attributes used as inputs of the LSTM network

| Attribute name | Type | Description | Units |
|---|---|---|---|
| P | Dynamic | catchment daily averaged precipitation | $mm \cdot d^{-1}$ |
| T | Dynamic | catchment daily averaged temperature | $^{\circ}C$ |
| peti | Dynamic | catchment daily averaged potential evapotranspiration | $mm \cdot d^{-1}$ |
| p_mean | Static | mean daily precipitation | $mm \cdot d^{-1}$ |
| pet_mean | Static | mean daily PET | $mm \cdot d^{-1}$ |
| aridity | Static | aridity | - |
| frac_snow | Static | fraction of precipitation falling as snow | - |
| high_prec_freq | Static | frequency of high-precipitation days | $d \cdot yr^{-1}$ |
| high_prec_dur | Static | average duration of high-precipitation events | $d$ |
| low_prec_freq | Static | frequency of dry days | $d \cdot yr^{-1}$ |
| low_prec_dur | Static | average duration of dry periods | $d$ |
| q_mean | Static | mean daily discharge | $mm \cdot d^{-1}$ |
| runoff_ratio | Static | runoff ratio | - |
| baseflow_index | Static | base flow index | - |
| area | Static | catchment area | $km^2$ |
| dpsbar | Static | catchment mean drainage path slope | $m \cdot km^{-1}$ |
| elev_mean | Static | catchment mean elevation | $m.a.s.l$ |
| urb_perc[1] | Static | percentage cover of suburban and urban | % |
| crop_perc[1] | Static | percentage cover of crops | % |
| inwater_perc[1] | Static | percentage cover of inland water | % |
| sand_prec[1] | Static | percentage sand | % |
| silt_prec[1] | Static | percentage silt | % |
| clay_prec[1] | Static | percentage clay | % |

[1] This attribute was not standardized by the global mean and standard deviation; instead, it was divided by 100. Dividing the value by 100 yields a number between 0 and 1. Since these values represent percentages, they can be interpreted directly, not just when compared to other basins.

## 2.5 Hybrid model: LSTM+SHM

Our hybrid model was created by combining a LSTM network, with the same architecture as the one from the previous section, with the SHM. The LSTM network predicts a set of values that serve as parameters for the SHM for each simulation time step. These parameter values are then utilized by the SHM to simulate the discharge, as indicated in Figure 3.

While the coupling of both models may appear straightforward from a general perspective, it is important to highlight several details. First, as one can notice from Figure 3, the forcing variables (P, T, ET) are used as inputs for both the LSTM and the SHM. The forcing variables and static attributes used as inputs for the LSTM are standardized using the method described in



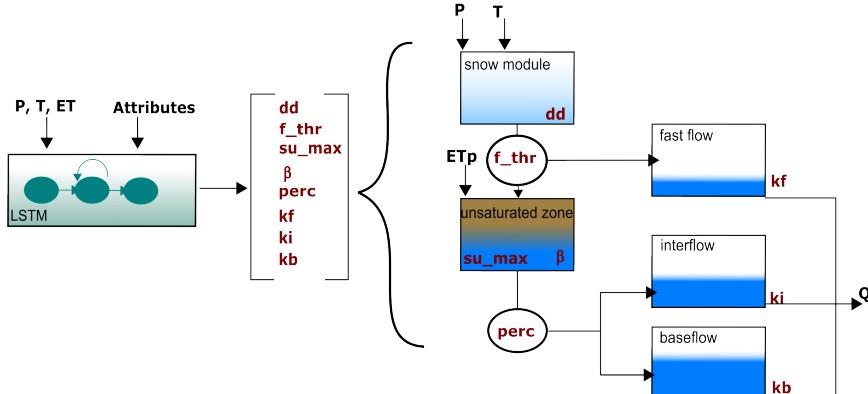

**Figure 3.** Structure of the hybrid hydrological model: LSTM+SHM

the previous section. However, due to the mass-conservative structure of the SHM, their input variables (P, T, ET) are used in the original scale. Second, it is important to consider that the SHM parameters have certain feasible ranges. While the LSTM

could theoretically learn these ranges, the optimization process becomes highly challenging due to the immense search space involved. We found that without constraining the parameter ranges, the LSTM was not able to identify parameters that yield a functional hydrological model. We hence predefined ranges within which the parameters can vary (see Table 2). These ranges were defined considering the findings in Beck et al. (2016) and Beck et al. (2020), which provide valuable insights into the appropriate parameter values of conceptual models. By defining these ranges, we not only reduce the computational costs of

the optimization but also ensure consistency with the methodology employed by Feng et al. (2022).

To map the output of the LSTM network to the predefined ranges, the $j$ outputs ($j = 8$, one per parameter) are passed through a sigmoid layer to transform the values to a [0, 1] interval. Then the transformed values are mapped to the predefined ranges through a min-max transformation, as exemplified in equation 2:

$$\theta_j = x_j^{min} + \text{sigmoid}(o_j) \cdot (x_j^{max} - x_j^{min}), \tag{2}$$

where $\theta_j$ is each of the values passed as parameters to the SHM, $o_j$ are the original outputs of the LSTM network, and $x_j^{min}$ and $x_j^{max}$ correspond to the minimum and maximum values of the predefined ranges (see Table 2) in which each parameter can vary, respectively.

It is important to highlight the difference in data-handling during training between the LSTM and hybrid model approaches. In the case of LSTM training, a batch size of $N = 256$ elements is selected from the pool of data consisting of the total number

of days multiplied by the number of basins. For instance, if we consider a training dataset spanning 10 years ($\sim 3652$ days) and encompassing 60 basins, we would have a total of $3652 \cdot 60 = 219\,120$ training points. From this pool, we randomly select 256 data points without replacement for each batch, run our model, calculate the loss function and update the network weights and biases based on this batch. To complete an epoch, we iterate through the $\frac{219120}{256} = 855$ batches. In the case of our hybrid model,





**Table 2.** Search range for SHM parameters during hybrid model optimization

| Parameter | Minimum value ($x_j^{min}$) | Maximum value ($x_j^{max}$) | Unit |
|---|---|---|---|
| $dd$ | 0.0 | 10.0 | $mm°C^{-1}d^{-1}$ |
| $f\_thr$ | 10.0 | 60.0 | $mm$ |
| $su\_max$ | 20.0 | 700.0 | $mm$ |
| $\beta$ | 1.0 | 6.0 | $-$ |
| $perc$ | 0.0 | 1.0 | $\%$ |
| $k_f$ | 1.0 | 20.0 | $d$ |
| $k_i$ | 1.0 | 100.0 | $d$ |
| $k_b$ | 10.0 | 1000.0 | $d$ |

$mm$ : millimeters, $°C$ : degree celsius, $d$ : days

the data handling differs for two specific reasons. First, the conceptual model component is designed to generate consecutive
predictions (the state of the model at time *t-1* is needed to make a prediction at *t*), demanding the selection of training instances
accordingly. Second, a warm-up period is necessary to stabilize the internal states of the SHM and prevent biased predictions
caused by incorrect initial conditions. To address these aspects, each training batch encompasses data from a single basin over a
2-year period, giving approximately 730 training points per batch. The initial year serves solely as a warm-up period (excluded
from the loss function), while the second year's data is utilized for actual training. Moreover, considering the regional nature
of the model, it is advisable to incorporate multiple basins when computing the loss function. To achieve this, we forward
multiple batches (e.g., 10) before calculating the loss and propagating the errors backward to update the weights.

To provide an example that illustrates the concepts discussed in the previous paragraph, let's consider the same scenario of a
10-year dataset with 60 basins. In this case, we can create nine 2-year windows (e.g., 1-2, 2-3, 3-4, ..., 9-10). For each window,
only the second year's data is utilized to calculate the loss, so no information is repeatedly used for training. Therefore,
a batch will be composed of a 2-year window for a specific basin, resulting in $N = 365 \cdot 2 = 730$ training points. Given
this configuration, the epoch containing the whole dataset is composed of $9 \cdot 60 = 540$ batches. Lastly, considering that we
backpropagate the errors and update the weights every 10 batches, we would have a total of $54$ updates per epoch.

## 3 Results and Discussion

### 3.1 LSTM vs LSTM+SHM

To address our research questions, the initial task was to develop our hybrid model (LSTM+SHM). By following the data
and methods outlined in Section 2, we achieved a performance comparable to that of an LSTM. When evaluated across the
60 testing basins, the LSTM yielded a median NSE of 0.84, while our hybrid model achieved 0.82. Figure 4 displays the
cumulative distribution function (CDF) of the NSE values, clearly demonstrating the close performance between both models.





In the same figure, we can see that both models outperformed the basin-wide calibration of the SHM model, which achieved

a median NSE of 0.72. Additionally, the performance of both models aligns with the reported median NSE of 0.88 in Lees et al. (2021) for their benchmark LSTM applied to the CAMELS-GB dataset. Despite the disparity in the testing datasets, the consistency and proximity of the results further validate the reliability of our findings.

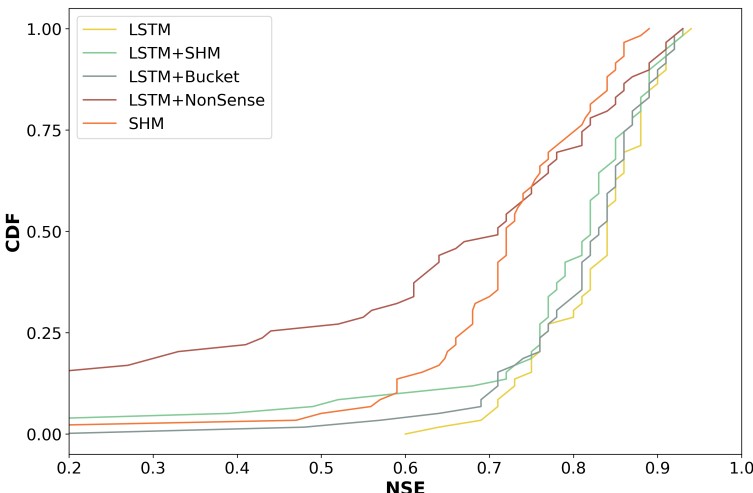

**Figure 4.** Cumulative density functions of the NSE for the different models

One point worth explaining is the decreased performance of our hybrid model compared to the LSTM in the low-performing basins. More specifically, there are 6 basins in which the hybrid model has lower performance than the lowest reported per-

formance by the LSTM across all basins. This drop in performance can be explained by two factors. First, the three lowest-performing basins in the hybrid model (ID 39010, 28015 and 43008) are also the lowest-performing in the SHM basin-wise calibration, which suggests a problem with the input data. The LSTM network has the capability to account for biases in the forcing variables (e.g. precipitation or evapotranspiration) because mass conservation is not enforced. However, both the conceptual and the hybrid approaches have a mass conservative structure, so the input quantities cannot be adjusted. This problem

was also reported by Feng et al. (2022) when applied to certain basins in CAMELS-US. The problem with the other three low-performance basins has to do with the training process. As detailed in Section 2, the hybrid model was trained using a regional approach that considers multiple basins together when calculating and backpropagating the loss. It was observed that these three basins fall within the range of the lowest observed high discharges in the database. Consequently, when computing the loss function for multiple basins, the relative contribution of these "dry" basins is diminished, and the optimization process

places a greater weight on achieving good performance in basins with higher discharges. In contrast, the training of the LSTM mitigates this issue to some extent by normalizing the input and output data in a preprocessing step. However, such normalization is not feasible for our hybrid model due to the constraints imposed by mass conservation. Feng et al. (2022) addressed this problem by modifying the structure of the loss function, but our experiments in that direction yielded only marginal variations.





It is important to highlight that this issue was observed in only a few basins, and in 90% of the basins, the performance of the
hybrid approach is fully comparable with the LSTM.

In summary, we observed comparable performance between the LSTM and LSTM+SHM models. Moreover, both models
outperformed the SHM-only, which indicates that the dynamic parameterization given by the LSTM is able to improve the
predictive capability of the model. This finding aligns with Feng et al. (2022), where they reached a similar conclusion despite
using a different conceptual model and applying it to a different dataset. It also coincides with Hoge et al. (2022) and Kraft
et al. (2022), who showed that fusing deep learning methods with hydrological mechanistic models allows reaching state-of-
the-art performance. Moreover, it conforms with Mendoza et al. (2015), who argues that fixed parameters can over-constrain a
model's ability to make good predictions. However, as described in the introduction, we are interested in looking at the LSTM-
SHM interaction, to evaluate if the good performance of the hybrid model is due to the right reasons and based on a consistent
interaction between the two model approaches, or if the LSTM network is overwriting the conceptual element. This will be
explored in the following section.

## 3.2    Effect of different regularizations

The first step to answer the aforementioned research question was to evaluate if the dynamic parameterization given by the
LSTM has the capability to overwrite the regularization imposed by the conceptual model. For this, we conducted two experi-
ments, in which the structure of the conceptual model was modified. In the first experiment (see Figure 5a), we substituted the
SHM with a single linear reservoir, leading to the removal of most hydrological processes typically represented in a conceptual
model through different reservoirs and interconnecting fluxes. A single bucket model only assures mass conservation and a
dissipative effect in which the input is lagged based on the recession coefficient in combination with a macroscopic storage. As
observed in Figure 5a, the model involves two calibration parameters: the recession parameter $k$ and a factor for the evapotran-
spiration term ($\alpha$). Similar to the previous cases, we defined predefined ranges in which the parameters were allowed to vary,
with k in the range [1 - 500] and $\alpha$ between [0 - 1.5]. The performance of this hybrid model (LSTM+Bucket) is fully compa-
rable to that of the LSTM and LSTM+SHM, achieving a median NSE of 0.83 (see Figure 4). For reference, when calibrated
without dynamic parameterization, the median NSE of the stand-alone bucket model drops to 0.55. This finding indicates that
the LSTM's dynamic parameterization effectively compensates for the missing processes, and the regularization provided by
the single bucket is insufficient to impact the model's performance.

Given the insights gained from the LSTM+Bucket experiment, we conducted a second experiment introducing an intention-
ally implausible structure in the conceptual model, referred to as LSTM+NonSense. As shown in Figure 5b, we removed the
fast flow reservoir, creating a single flow path comprising the baseflow, interflow, and unsaturated zone, in that specific order.
We also maintained the parameter ranges specified in Table 2, which restrict the baseflow reservoir to have smaller recession
times than the interflow. Then, only after the water has been routed through these two reservoirs it can enter the unsaturated
zone, where the outflow is no longer controlled by a recession parameter but by an exponential relationship depending on
*su_max* and $\beta$. Under static parameterization, the stand-alone NonSense model yielded a median NSE of 0.57, really similar
to the performance of the stand-alone bucket model. However, after applying dynamic parameterization, the LSTM+NonSense





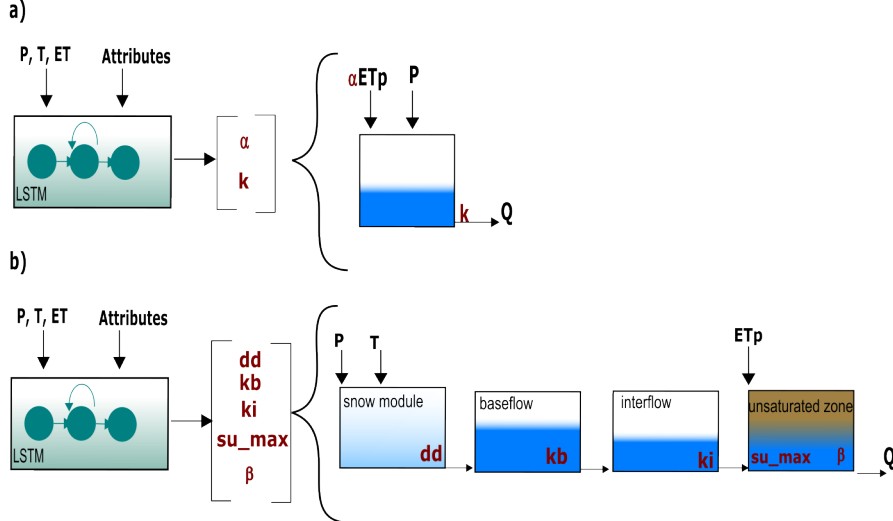

**Figure 5.** Structure of the different regularization: a) LSTM+Bucket b) LSTM+NonSense

achieved a median NSE of 0.71, similar to the original SHM model (see Figure 4). During the test we observed that the optimization routines tried to reduce the recession parameter of the baseflow and interflow, to avoid the initial lagging. This caused

the optimized parameters to reach the lower limits, which might have limited an additional performance increase. Expanding the parameter ranges might lead to a further performance gain, however, this would come at the cost of reducing the differences between the reservoirs, which contradicts the objective of the experiment. Taken together, these experiments provided valuable insights into addressing the first research question posed in the introduction: Can conceptual models effectively serve as a regularization mechanism for the dynamic parameterization given by the LSTMs? Based on our results, we observed that the

regularization offered by the conceptual model is not strong enough to reduce the hybrid model performance, and the dynamic parameterization given by the LSTM can even compensate for missing processes and implausible structures. Figure 6 highlights that, for some basins, we obtained a similar hydrograph for all models used. Therefore, we recommend being careful about using this hybrid scheme for comparing different types of conceptual models or multiple working hypotheses (Clark et al., 2011), especially if we are evaluating model adequacy by performance alone, as the overall performance can be adjusted

by the data-driven part.

However, the fact that the data-driven component possesses this capability does not necessarily imply that a well-structured conceptual model cannot be consistently utilized by the LSTM. Our next objective is to further explore this hypothesis, addressing the second research question, as presented in the following section.

## 3.3 Analysis of LSTM+SHM

In order to assess the interpretability of the conceptual part of our LSTM+SHM model, we conducted several tests. Hybrid models, as highlighted by Feng et al. (2022), Kraft et al. (2022) and Hoge et al. (2022), offer the advantage of providing access



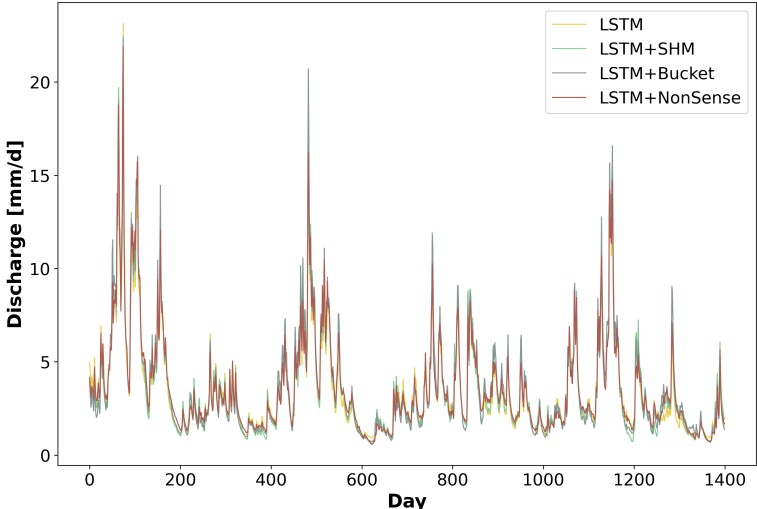

**Figure 6.** Specific discharge series in the testing period for basin ID 15006, simulated by the different models

to untrained variables as the model's states and fluxes have dimensions and semantic meaning. As such, our first test aimed to compare the model's states with external data. Specifically, we evaluated the filling level of the unsaturated zone reservoir, representing soil moisture in our model (LSTM+SHM), against ERA5-LAND soil water volume information. The process of utilizing reanalysis data and the necessary data processing steps for this comparison are detailed in Section 2.2.

310 Across the 60 basins in the testing period, the LSTM+SHM model demonstrated a median correlation of 0.83 when compared against the soil moisture simulation provided by ERA5-LAND. This result indicates that the unsaturated zone dynamics are well represented in our model and that the hybrid approach allows us to recover this variable without including any soil moisture information in our training. Moreover, the correlation achieved by the LSTM+SHM model (0.83) surpassed that of

315 the LSTM+Bucket (0.74) and the LSTM+NonSense (0.59), suggesting that our LSTM+SHM model effectively utilizes the well-structured conceptual part to get better predictions of the untrained variables. It is important to note that the correlation obtained by the LSTM+Bucket and LSTM+NonSense models are not insignificant, which can be attributed to the strong dependence of soil moisture with the precipitation and evapotranspiration series, both of which serve as boundary conditions for all models.

320 In addition to the comparison with external data, we also examined the correlation between soil moisture estimates produced by the LSTM+SHM model and the stand-alone SHM. The median correlation value of 0.96 further confirms that the unsaturated zone within our hybrid model operates in accordance with our initial expectations. Figure 7 exemplifies this agreement for basin 11001, where the modelled (LSTM+SHM) and ERA5-LAND (swvl3) series exhibit a correlation of 0.83, equivalent to the median correlation observed across all 60 basins for our LSTM+SHM model. For reference, the median correlation of the

325 stand-alone SHM over the 60 basins was 0.86





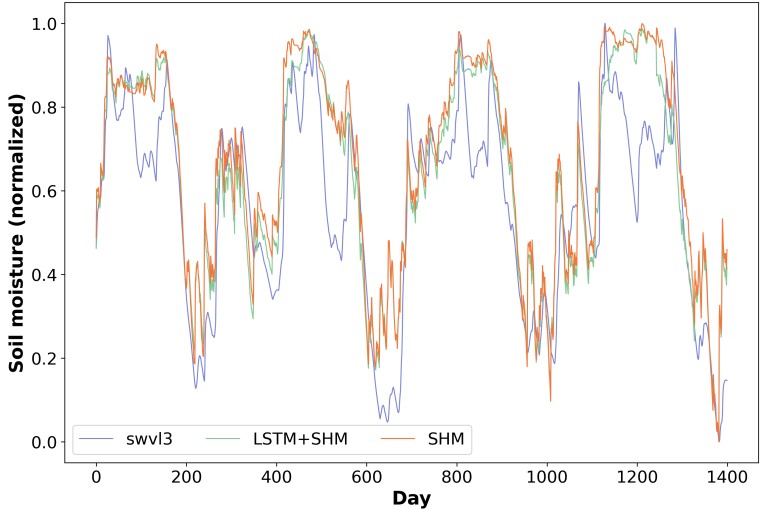

**Figure 7.** Soil moisture time series comparison during the testing period for basin ID 11001

Expanding on the comparison between LSTM+SHM and SHM, we also compared the proportions of discharge originating from the fastflow, interflow, and baseflow reservoirs for both the LSTM+SHM and the SHM models. To determine the overall proportion, we calculated the percentage of total outflow, during the testing period, attributed to each reservoir for each basin, and then averaged the results across the 60 basins. For the hybrid model, we found that the fastflow, interflow, and baseflow

reservoirs contributed to 14%, 59%, and 27% of the total outflow, respectively. These values are in a similar range with the corresponding proportions reported by the SHM model, which were 3%, 66%, and 31%, respectively. This experiment further illustrates the strong agreement between both models.

In order to further evaluate the consistency of our hybrid model, we conducted a final experiment analyzing the parameter variation over time. Figure 8 presents the results for four calibration parameters: $su\_max$, $\beta$, $kb$, and $ki$, across two different

basins (reasons for choosing these two basins are explained below). We begin by examining the behaviour of the first two parameters.

The purpose of $su\_max$ and $\beta$ is to control the water transfer from the unsaturated zone reservoir to the interflow and baseflow, following equation 3:

$$qu\_out = qu\_in \cdot \left( \frac{su}{su\_max} \right)^{\beta}, \tag{3}$$

where $qu\_out$ represents the water going out of the unsaturated zone, $qu\_in$ represents the water entering the unsaturated zone from precipitation and snowmelt, $su$ refers to the unsaturated zone storage or soil moisture, and $su\_max$ and $\beta$ are the calibration parameters. It is important to note that the value of $su$ cannot exceed $su\_max$, which forces their quotient to be less or equal to one. Consequently, a larger value of $su\_max$ and/or $\beta$, leads to a decrease in the unsaturated zone outflow.

The parameter variation in basin id 10003 presents clear seasonal patterns that align with our initial expectations. During

low-flow periods, both parameters increase, resulting in reduced water availability for the remaining two reservoirs, and,





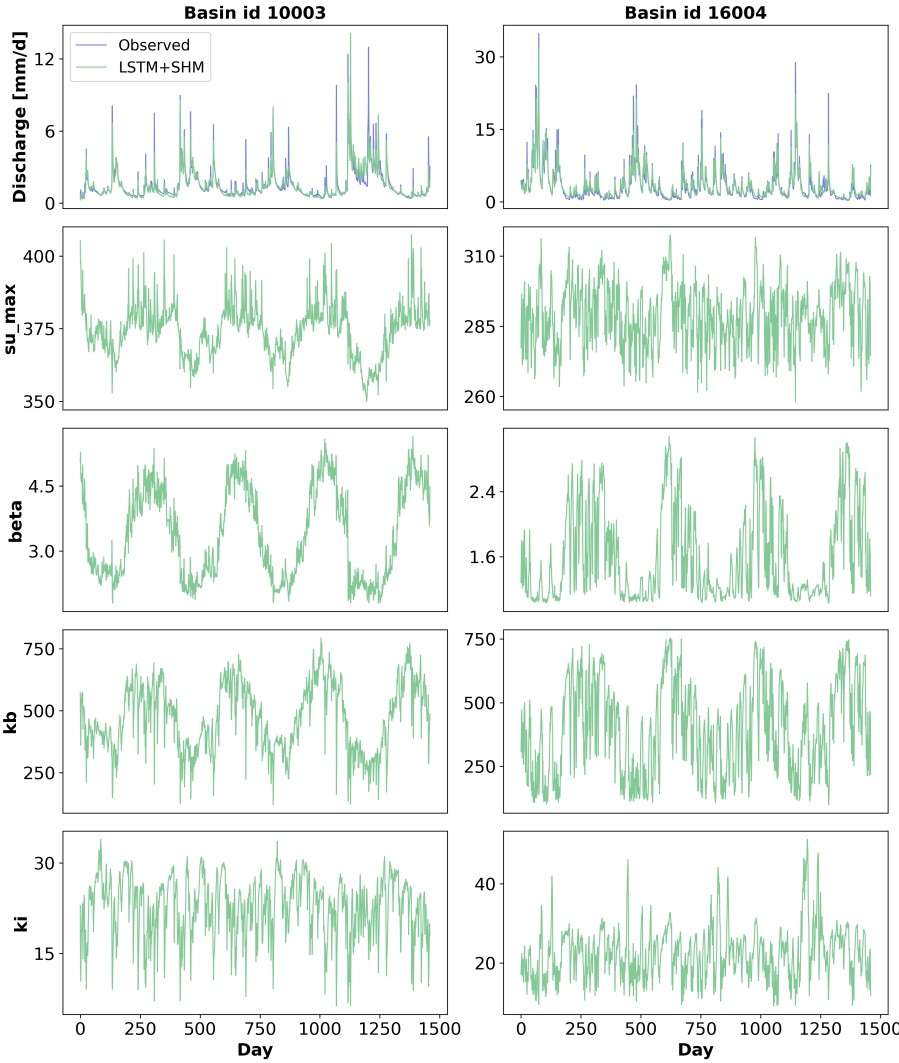

**Figure 8.** Time variation of parameters for basins 10003 (left column) and 16004 (right column). It should be noted that the Y-Axis ranges of the two basins differ.

consequently, a decrease in the total outflow. On the other hand, during high-flow periods, the opposite happens. As both parameters decrease, there is an increase in water availability, resulting in higher outflows. Basin 10003 was selected for this example due to its reported NSE of 0.83, which closely aligns with the median value of 0.82 obtained for the 60 basins.

Even though similar patterns were observed in other basins, not all basins displayed such behavior. The results for basin

16004 in the right column of Figure 8 show noisier behaviour with less clear patterns, particularly for *su_max*. However, despite the parameter noise, the observed and simulated discharge matched well, resulting in an above-average NSE of 0.89. These good results, despite the noisy parameters, may indicate cases of cross-compensation, where the LSTM's flexibility





allows for one parameter to compensate for another, causing a different model behaviour than expected, but still getting good results. This suggests that, for some basins, the regularization provided by the SHM is not fully effective. Nonetheless, the general agreement between the LSTM+SHM and SHM unsaturated zone reservoir (median correlation of 0.96) and interflow reservoir (median correlation of 0.74) indicate a strong agreement between both models.

Regarding the other two parameters, $kb$, and $ki$ have a linear relationship with their respective outflows, acting as the denominator of their storage units ($q_{[i,b]} = \frac{s_{[i,b]}}{k_{[i,b]}}$). For $kb$, we can observe seasonal patterns, which allows the model to further increase the baseflow in wet periods and reduce it during dry seasons. This also aligns with our knowledge that hydraulic conductivity is lower when the soil is drier. On the contrary, $ki$ displays faster variations for both basins. We argue that this response is due to the fact that the interflow storage is reacting to information pulses on a faster scale, so the recession parameter varies accordingly.

The oscillations in $ki$ and $kb$ also indicate structural deficiencies in the SHM, which the LSTM is attempting to compensate for. As discussed in Shen et al. (2023), process parameterizations are subjected to high uncertainty due to, among other factors, the coarse scale of the models. In our case, we are using the SHM (both as SHM-only and LSTM+SHM) in a lumped form, which means that one SHM is being used to represent catchment-scale processes. Consequently, we are encompassing multiple uncertainties and subprocesses into a single parameter, which our LSTM varies to increase the performance. An alternative approach is to increase the complexity of our process-based part, reducing the necessity of the data-driven method to compensate for structural deficiencies. For example, Feng et al. (2022) represent the catchment processes using 16 HBVs acting in parallel, which are parameterised through a LSTM. In their case, the recession parameters were predicted as constant in time, and the necessary flexibility to get state-of-the-art performance and account for missing sub-processes was considered by the semi-distributed format.

The main objective of this section was to answer the research question: Does the dynamic parameterization of the data-driven component overwrite the physical interpretability of the conceptual model? Therefore, to better examine the results, we intentionally granted high flexibility to the dynamic parameterization. Based on the experiment's outcome, we argue that the conceptual part of the hybrid model still maintains good interpretability. However, we can see that there are cases of parameter cross-compensation and compensation for missing processes, which are not ideal. To deal with this, techniques to increase the LSTM regularization can be used. For example, one may use conceptual or physical-based models that account for more processes, to reduce the necessity to overcompensate for missing processes in order to boost performance. One can also limit the time-to-time maximum variation that the LSTM is allowed to impose in the parameters, which would lead to less noisy behaviour.

## 4 Summary and Conclusions

In recent years, data-driven techniques, specifically LSTM, have outperformed conceptual hydrological models for rainfall-runoff prediction. Despite notable advancements in linking hydrological concepts to the internal states of the LSTM, concerns towards their interpretability and robustness still persists. Because of this, the idea of creating hybrid models by combining





data-driven techniques and conceptual models has arisen as a feasible mid-ground, where the improved performance of data-driven techniques is combined with the interpretability of physical models. Following this line of thought, Feng et al. (2022) used as a hybrid approach the parameterization of a hydrological model by an LSTM network. The authors demonstrated the potential of the technique to achieve comparable performance as purely data-driven techniques and outperform stand-alone

conceptual models. Kraft et al. (2022) also achieved promising results following a similar process.

Motivated by this outcome, our article dug into the effect of dynamic parameterization in our conceptual model, and the consequences this might have on the interpretability of the model. More specifically, we tried to answer the questions: 1. Do conceptual models effectively serve as a regularization mechanism for the dynamic parameterization given by the LSTMs? 2. Does the dynamic parameterization of the data-driven component overwrite the physical interpretability of the conceptual

model?

The first step towards answering these questions was to create a hybrid model. We coupled an LSTM network with a conceptual hydrological model (SHM), using the former as a dynamic parameterization of the latter. In our study, we demonstrated that our hybrid approach (LSTM+SHM) was able to achieve state-of-the-art performance, comparable to purely data-driven techniques (LSTM). Both models were trained in a regional context, utilizing a subset of the CAMELS-GB dataset. The me-

dian NSE obtained for the hybrid case was 0.82, while the LSTM achieved 0.84. Both models outperformed the basin-wise calibrated conceptual model, which served as the baseline and achieved a median NSE of 0.72. These findings align with existing literature. For instance, Feng et al. (2022) reached similar conclusions when applying a hybrid model to the CAMELS-US dataset. The consistent results across different hybrid model structures and datasets highlight the robustness of the method.

Having accomplished a well-performing hybrid model, we addressed the first research question. By modifying the regular-

ization given by the conceptual model, we tested to which degree the dynamic parameterization given by the LSTM has the potential to compensate for missing processes. We proved that a hybrid model composed of an LSTM plus a single bucket (LSTM+Bucket) was able to achieve the same performance as the LSTM+SHM and LSTM-only. This indicates that the regularization given by the conceptual model is not strong enough to drop the predictive capability of the hybrid model, and missing processes can be outsourced to the data-driven part. We also demonstrated that if we use an intentionally implausible structure

(LSTM+NonSense), the LSTM had the flexibility to artificially increase the performance. Therefore, we recommend being careful about using this hybrid scheme for comparing different types of conceptual models, especially if we are evaluating model adequacy by performance alone, as the overall performance can be adjusted by the data-driven part.

However, the fact that the data-driven component possesses this capability does not necessarily imply that a well-structured conceptual model cannot be consistently utilized by the LSTM. Therefore, we further analyzed the internal functioning of our

LSTM+SHM model, in order to answer our second research question. We started by comparing the soil moisture predicted by our hybrid model with ERA5-LAND data. This test addressed one of the main benefits of hybrid models over purely data-driven ones, which is their ability to predict untrained variables. Across our testing set, comprised of 60 basins, we obtained a median correlation of 0.83 between our simulated soil moisture and the ERA5-LAND data. This result indicates that our hybrid model was able to produce coherent temporal patterns of the untrained variable, without having access to the corresponding data

during the testing period. We also compared the unsaturated zone reservoir of the LSTM+SHM against the unsaturated zone



of the stand-alone SHM, which reported a median correlation of 0.96. These results indicate that the dynamic parameterization was operating the unsaturated zone reservoir in a consistent manner and according to our initial expectations.

The last section of the study presented the results of the dynamic parameterization for two basins. In one basin, the parameters had a cyclic seasonal behaviour, operating the SHM model consistently with the low-flow and high-flow periods. In the other
basin, even though these behaviours were present, there was an increased level of noise. We argue that these cases may indicate a certain degree of cross-compensation, where the LSTM's flexibility allows for one parameter to compensate for another. This is an indication that a stronger regularization might be beneficial in some cases.

*Code availability.* The codes used to conduct all the analyses in this paper are publicly available at: https://github.com/KIT-HYD/Hy2DL and https://doi.org/10.5281/zenodo.8289021

*Data availability.* The data used to conduct all the analyses in this paper is publicly available at: https://doi.org/10.5281/zenodo.8289409

*Author contributions.* The original idea of the manuscript was developed by all authors. The codes were written by E.A.E with support from R.L and M.A.Ch. The simulations were conducted by E.A.E. Results were further discussed by all authors. The draft of the manuscript was prepared by E.A.E. Reviewing and editing was provided by all authors. Funding was acquired by U.E and N.B. All authors have read and agreed to the current version of the manuscript.

*Competing interests.* Some authors are members of the editorial board of HESS.

*Acknowledgements.* We would like to thank Dr. Marvin Höge and M.Sc María Fernanda Morales Oreamuno for their valuable input.

*Financial support.* This project has received funding from the KIT Center for Mathematics in Sciences, Engineering and Economics under the seed funding program. The article processing charges for this open-access publication were covered by the Karlsruhe Institute of Technology (KIT).



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
