# Peer review of "To Bucket or not to Bucket? Analyzing the performance and interpretability of hybrid hydrological models with dynamic parameterization"

_EGUsphere, 2023_

## Referee Comment (RC1)

Manuscript: To Bucket or not to Bucket? Analyzing the performance and interpretability of hybrid hydrological models with dynamic parameterization.

*General Comments*

The authors introduce and analyse a hybrid hydrological model consisting of a conceptual hydrological model and a LSTM data-driven model to estimate time dependent model parameter dependent on the same inputs as used to drive the conceptual model. The intension is to keep the excellent performance of data driven approaches that have been demonstrated in recent years, but also to keep or improve the interpretability of such data driven approaches.

In general, I am in favour of an intensive analysis of such approaches, and think the manuscript is well suited for the readership of HESS, in continuation of a significant number of important papers in this area in the same journal.

It is in general well written and figures support the understanding and flow of the text! However, I have a number of major and minor comments/suggestion that I believe would improve the manuscript and should be addressed before final publication.

- The authors motivate they work by a paper of Feng et al. who propose a general framework of hybrid dPL modelling. They use the HBV model as a basis and estimate static and dynamically HBV parameters using Catchment parameters and meteorological input (as used do force HBV).  This paper extends and slightly varies the this approach by analysing simple bucket based models as well as (what they call) NonSense model. Dynamic parameters are estimated with an LSTM DL. Research question 1 is "do conceptual models serve as a regionalization mechanism for thwe dynamic parameterization? I do think this is an important question (and I miss the reference of Frame et al, 2022 in this context), however, I believe it is not addressed in such a rigorous way as would be needed here. Conceptual models can range over a large range of complexity. Wha,t if we would just apply a simple equation relating   Rainfall to runoff (Q = c(x,t) * P) and allow c to be estimated by a LSTM as suggested. This is the simplest model I can think of, and then I would systematically increase the complexity of the conceptual models.
  (Frame, J. M., Kratzert, F., Klotz, D., Gauch, M., Shalev, G., Gilon, O., Qualls, L. M., Gupta, H. V., and Nearing, G. S.: Deep learning rainfall–runoff predictions of extreme events, Hydrol. Earth Syst. Sci., 26, 3377–3392, https://doi.org/10.5194/hess-26-3377-2022, 2022.)
  In that procedure I would suggest to use a much wider set of catchments and characteristics in order to see under what physio-geographical properties and climate conditions (as has been used of plenty other previous application) to answer research question 1 in a more general way!
- Research question 2 addresses the physical interpretability of conceptual models and whether it is comprised by data driven dynamic parameterization. Fig. 8 shows some of the parameters for 2 catchments and how they vary in time. I am missing a few points that should be discussed: i) Are the variations of parameters du to structural imitations of the conceptual model component, or is it just needed because of averaging non-linear processes over spatial variable catchment characteristics, or is it compensating for biases in the ERA5

input data? Or all three? What do I learn from Fig. 8? Which weight is assigned to each individual input for driving the variation? ii) How does the methodology compare to "more classical/statistical approaches" such as state and time dependent parameter estimation techniques. iii) How does the methodology compare in philosophy and potential to approaches that have been introduced by e.g. Feigl et al. (2022), what do we learn here in this approach from mistakes?

(Feigl et al., 2022, Learning from mistakes-Assessing the performance and uncertainty in process-based models. Hydrological Processes 36).

- Overall, I miss a kind of "surprise" concerning the analysis – could that be more emphazided.

Specific/technical Comments

The following minor comments/suggestions I would like to make:

- L9ff: The last part of the abstract is hard to understand/follow – I read it before the rest of text and did not know what is meant.
- L20: Reference needed.
- L136: how is ETp calculated (may one short sentence)
- L161: how you calculate the gradiants for if/then and iterative loops with state updates?
- L214: is 855 batches true hen you consider tat one data point consideres 180 previous days as input?
- L216: Why not optimizing the initial conditions?
- L232: this refers to one major comment – when is the model complex enough so that the LSTM is able to produce the full output space just by varying parameters!? Is this already possible with the structure I suggested). When an I see limitations/restictions?
- L265: what is the criterium for overfitting! Have you used ensembles of optimized networks to see how robust results are?
- Fig. 6: it is hard to see any differences, perhaps you can enlarge an interesting part of the time seies!
- L309: I would guess that ERA5-Land data are also computed and not observed quantities. So it is a model state intercomparison!
- L329: why looking at average values and not show the distribution?
- L385: what has this paper contributed to a better understanding in this context! Be specific!
- L402: What s new compared to Feng et al., what are different findings!
- L417: States (instead of variables!?
- L421: correlation is a very weak measure-of -goodness-of-fit especially when dealing with cyclic data/processes)

Overall, I feel, the manuscript has in general the potential to be a valuable contribution to HESS, however, questions and issues raised in the general comments would need to be addressed and discussed to a significant part before final acceptance.

---

## Referee Comment (RC2)

**Review of HESS Manuscript Titled "To Bucket or not to Bucket? Analyzing the performance and interpretability of hybrid hydrological models with dynamic parameterization"**

Discussion Paper: https://doi.org/10.5194/egusphere-2023-1980

Reviewer:  Grey Nearing

**Summary of Paper**

The paper addresses the following two questions:

1. Do conceptual models serve as an effective regularization mechanism for the dynamic parameterization of LSTMs?

2. Does the data-driven dynamic parameterization compromise the physical interpretability of the conceptual model?

Neither question is answered directly in the paper, but as far as I can tell, the authors intend to convey that the answer to both questions is "No".

Additionally, the authors conclude that while it does not help increase model skill to add a bucket model to an LSTM, doing so does allow the model to predict variables other than the training target (which here is streamflow).

**Summary of Review**

In general I think these are interesting and informative experiments. Also, I want to express how refreshing it is to review an ML-based hydrology paper that uses best practices. Specifically, the authors are careful to train the models in a way that matches prior publications, and they build on existing community benchmarks (although some improvement on using the benchmarks more carefully would help).

I have two questions/concerns with the conclusions:

1) The lesson that I take from the quantitative results in this paper is that there is no value in adding bucket models to LSTMs. In other words, there appears to be a clear and direct answer to the question in the title. Bucket models do not increase skill, and there are no examples in this paper of something that the hybrid modeling structure can predict that the LSTM alone cannot. It appears that the bucket model (unless intentionally configured to be nonsense) is a transparent "head layer" in the deep learning model stack that contributes no information.

I suspect that the authors might respond to my interpretation of their results by saying that the bucket models allow for estimating non-target states and fluxes. However, what is conspicuously missing from this paper is a quantitative benchmark against the results from Lees er al (2022), who show that the LSTM can estimate (unobserved) soil moisture and snowpack (the two non-target variables explored here).

I'm searching for anything in this paper that provides evidence that there is any value to using bucket models. Otherwise, I suggest that the authors answer the titular question directly.

2) I'm not 100% sure that I understand the reasoning behind concluding that the bucket models cannot regularize the LSTM. The 0.71 median NSE from the LSTM+NonSense model seems to indicate that this regularization is possible, in principle. Interpreting the bucket models as head layers on the LSTM, this experiment seems to indicate that a really bad head layer can result in information loss (which makes sense to me).

I am not 100% sure what is going on in these experiments, but here is one hypothesis. Bucket models (I would prefer to think of them as head layers, because the lesson generalizes beyond bucket models) can eat information (presumably by poorly regularizing the loss response), but "reasonable" bucket models used as head layers don't add any information. Whether or not the "reasonable" bucket models are regularizing the loss response surface is unknown, since it appears that the LSTM is finding a similar solution either way. However, since the LSTM is finding similar solutions either way – there is no useful regularization from these head layers, and also no information loss.

I would suggest that there are probably ways that we could look more closely at the functionals that the LSTM and hybrid models discover. Are they really finding similar relationships? Sensitivity analyses, input/output response surfaces, integrated gradients, etc. I'm not sure what the best approach would be (one of many would probably work), but I think that if you wanted, you could be more rigorous about understanding how these head layers are interacting with the LSTM component of the model to create functional mappings. I don't really care whether the authors do that for this paper or not.

**Specific Comments**

Line 155: "...selected the one that performed the best for each basin." Was this selection done with train-period, evaluation-period, or test-period data?

Line 215: I'm not sure that I understand (or agree with) the distinction being made in this paragraph. The LSTM has a state vector, just like the SHM – there is no difference here. The LSTM requires consecutive predictions in exactly the same way as SHM – again, there is no difference. We use a sequence-to-one training procedure when training the LSTM because this provides more diversity in the minibatch and accelerates training. You could do exactly the same thing for training SHM and/or the hybrid model. You are using a 2-year period for the hybrid model, and you could accomplish the same thing by using a 2-year sequence length for the hybrid model while still training sequence-to-one. You can think about the LSTM as having a 180-day spinup while the hybrid model has a 365 day spinup, and the LSTM is trained seq2one while the hybrid model is trained seq2seq. It's fine if you've found that training the hybrid model using a sequence-to-sequence approach is better, but the way that is motivated (and the way this distinction is framed) in this paragraph is not correct.

Line 230: Do these median statistics come from ensembles (as is used by Kratzert et al), or from single LSTM/Hybrid models?

Line 235: I think it is too strong to claim that the "consistency and proximity" with Lees "validate the reliability of findings." This isn't a rigorous way to build on a community benchmark. A better approach is to actually recreate the existing benchmark exactly, which demonstrates that the models are built and trained correctly, then transition the experiment to the basins / time periods that you want to use for this study. This is, for example, what we did in the Frame et al. papers that required different training/test sets than what were used by previous community benchmarks.

Line 240: " The LSTM network has the capability to account for biases in the forcing variables (e.g. precipitation or evapotranspiration) because mass conservation is not enforced." We have a paper that demonstrates this explicitly:

> Frame, Jonathan M., et al. "On strictly enforced mass conservation constraints for modelling the Rainfall‑Runoff process." *Hydrological Processes* 37.3 (2023): e14847.

Line 270: What does the word "overwrite" mean in this sentence? I think that being very specific here about what you want to test is important because how the bucket model influences the LSTM is the central question. There is no such thing as "overwriting". I suggest being crystal clear and precise about what you are envisioning when you ask this question.

Line 300: I agree that the results from section 3.2 don't indicate that a conceptual model can't be used, but again, I want *some* type of evidence that this has value. I don't care about subjective arguments or what members of "the community" might think (e.g., line 35), I want some type of real, quantitative, scientific evidence that this is a useful thing to do. Showing strong correlation with soil moisture is interesting, but isn't anything new for these ML-based rainfall-runoff models.

Line 315: "...suggesting that our model effectively utilizes the well-structured conceptual part to get better predictions of the untrained variables." I disagree – this is one possible explanation for these results, but there is another possibility (that SHM is simply doing nothing), and these experiments can't differentiate between these two possibilities.. These results show that the poorly-structured head layers cause the LSTM to lose information about soil moisture, but there is no indication that the "well-strucutred" model is being used in any way. The LSTM could be (and I suspect is) providing all of the information about soil moisture here, and the SHM model is doing nothing. A way to test this would be to use the Lees methodology and see what the LSTM can do alone to predict soil moisture. You probably will need to check both the LSTM standalone and the LSTM component of the hybrid model, since the method by which SHM would add value would be through regularization during training, and it is likely (although not guaranteed by the data processing theorem, due to the meteorological inputs to SHM) that the LSTM in the hybrid model will contain all or most of the information about soil moisture that is present in that whole model.

Line 345: The su_max and beta parameters being higher *might* result in lower water availability for down-stream buckets, but really what they are doing is reducing the outflow *rates*. More total water could compensate for this. Is all that is happening here is that the model is pulling a lever to reduce the total output, or is it more complicated – does the rate need to reduce for some reason. What do total inputs (P - ET) look like over these seasonal cycles? If P - ET is lower for the low-flow seasons, why is it necessary that the flow rates from the unsaturated zone also must be lower? Anyway, it's a little simplistic to just say that higher parameter values result in lower water availability in the downstream buckets.

Line 350: Is there any reason to believe that "noisy" parameter values indicate parameter interaction? Could it just be that there is no low-frequency signal that is needed in this basin to compensate for lack of information in seasonal precipitation signal? Also, what does "noisy" mean? Why do we think this high-frequency variability is noise?

This whole set of experiments is leading me to hypothesize that the problem with the bucket models is that there aren't enough buckets. More buckets allow for more flexibility in the mixing of residence times. We could use hundreds of buckets wired in parallel and series, with skip connections (to account for mass transfers between buckets on timescales shorter than the number of timesteps represented by the distance between two buckets in series). We could train this much bigger bucket model and look at residence time mixing ratios over seasons. And really, this would not be very much different than using a set of fully connected layers with linear coefficients, which – surprise, surprise – is exactly what the head layer on the standalone LSTM is.

Line 415: Referring back to my main criticism, I would like to take issue with this sentence from the conclusion section: *"This test addressed one of the main benefits of hybrid models over purely data-driven ones, which is their ability to predict untrained variables."* I do not believe that it has been shown that this is a benefit of the hybrid modeling approach. LSTMs do this by themselves, and the authors even cited papers in their introduction that demonstrated this. I can

imagine thinking that this might be a benefit, but that was not demonstrated in this paper (or any other that is currently published).

One question I have after reading all this is about why the parameters should change over time (i.e., why dynamic parameters work and static ones don't). Of course, the answer appears to be simply that the bucket model isn't providing any information and since the LSTM is doing the predicting, of course that prediction needs to be dynamic. But I wonder whether the dynamics in the conductivity parameters in particular might vary with moisture content in a some way that could be interpreted as a characteristic curve?

---

## Author Comment (AC2)

**Response to RC2: Comment of eguspere-2023-1980. Referee Grey Nearing. 13 Nov 2023**

**We want to thank the referee for the detailed evaluation of our paper. In this document we answer the questions, comments and suggestions given. We will address those comments individually. For clarity, the original comments posted by the referee are written in italic, while our answers are written in bold.**

*Summary of Paper*

*The paper addresses the following two questions:*

1. *Do conceptual models serve as an effective regularization mechanism for the dynamic parameterization of LSTMs?*
2. *Does the data-driven dynamic parameterization compromise the physical interpretability of the conceptual model?*

*Neither question is answered directly in the paper, but as far as I can tell, the authors intend to convey that the answer to both questions is "No".*

*Additionally, the authors conclude that while it does not help increase model skill to add a bucket model to an LSTM, doing so does allow the model to predict variables other than the training target (which here is streamflow).*

**Summary of Review**

*In general I think these are interesting and informative experiments. Also, I want to express how refreshing it is to review an ML-based hydrology paper that uses best practices. Specifically, the authors are careful to train the models in a way that matches prior publications, and they build on existing community benchmarks (although some improvement on using the benchmarks more carefully would help).*

**We thank the referee for the well-structured summary of our paper, and for the detailed revision he performed. His suggestions and comments were helpful to clarify concepts shown in the article.**

*I have two questions/concerns with the conclusions:*

1. *The lesson that I take from the quantitative results in this paper is that there is no value in adding bucket models to LSTMs. In other words, there appears to be a clear and direct answer to the question in the title. Bucket models do not increase skill, and there are no examples in this paper of something that the hybrid modeling structure can predict that the LSTM alone cannot. It appears that the bucket model (unless intentionally configured to be nonsense) is a transparent "head layer" in the deep learning model stack that contributes no information. I suspect that the authors might respond to my interpretation of their results by saying that the bucket models allow for estimating non-target states and fluxes. However, what is conspicuously missing from this paper is a quantitative benchmark against the results from Lees et al (2022), who show that the LSTM can estimate (unobserved) soil moisture and snowpack (the two non-target variables explored here). I'm searching for anything in this paper that provides evidence that there is any value to using bucket models. Otherwise, I suggest that the authors answer the titular question directly.*

The idea of seeing the conceptual model as a head layer is quite accurate and is an alternative way of interpreting the hybrid model architecture we are using. We want to thank the referee for pointing this out. We will add the following table to the article to illustrate his idea and update our model description and discussion accordingly.

| Model | | Head | |
|---|---|---|---|
| LSTM | Input ->LSTM -> | Dense | -> Q |
| LSTM+Bucket | Input ->LSTM -> | Dense -> Bucket | -> Q |
| LSTM+SHM | Input ->LSTM -> | Dense -> SHM | -> Q |
| LSTM+NonSense | Input ->LSTM -> | Dense -> NonSense | -> Q |

In the experiments we conducted, and as stated in the paper, we were not able to increase the performance by adding the conceptual part. In future work we want to test if hybrid models present other advantages for different conditions (extrapolation), but until this point the remaining advantage that we see is the access to non-target states. We will explicitly include these statements in a revised version of the manuscript.

The main difference with Lees et al (2022) is that they need to train an additional model (which they call probe) to extract the non-target states, while in the method we propose no extra training needs to be done. In other words, the SHM is acting both as a head layer and a probe. The fact that in Less et al (2022) the probe can be as simple as a linear model, which requires few points to train, is not being argued, and in many cases, this will reduce the advantage given by our hybrid approach. However, we argue that there is a methodological difference between the approaches, as in their case another model needs to be trained to extract non-target states while in our case the architecture of the conceptual model takes care of this.

To make a more transparent comparison between these two methods, in a revised version of the manuscript, we will include the previous discussion about the difference between them. We will also include a comparison between our predicted soil moisture and the predicted soil moisture presented by Lees et al (2022).

2. *I'm not 100% sure that I understand the reasoning behind concluding that the bucket models cannot regularize the LSTM. The 0.71 median NSE from the LSTM+NonSense model seems to indicate that this regularization is possible, in principle. Interpreting the bucket models as head layers on the LSTM, this experiment seems to indicate that a really bad head layer can result in information loss (which makes sense to me). I am not 100% sure what is going on in these experiments, but here is one hypothesis. Bucket models (I would prefer to think of them as head layers, because the lesson generalizes beyond bucket models) can eat information (presumably by poorly regularizing the loss response), but "reasonable" bucket models used as head layers don't add any information. Whether or not the "reasonable" bucket models are regularizing the loss response surface is unknown, since it appears that the LSTM is finding a similar solution either way. However, since the LSTM is finding similar solutions either way – there is no useful regularization from these head layers, and also no information loss. I would suggest that there are probably ways that we could look more closely at the functionals that the LSTM and hybrid models discover. Are they really finding similar relationships? Sensitivity analyses, input/output response surfaces, integrated gradients, etc. I'm not sure what the best approach would be (one of many*

*would probably work), but I think that if you wanted, you could be more rigorous about understanding how these head layers are interacting with the LSTM component of the model to create functional mappings. I don't really care whether the authors do that for this paper or not.*

**By regularization, we are referring to adding extra information to our model that constrain the space of solutions. For example, an L1 regularization includes information that the solution should be sparse. Moreover, a specific loss function might include information that the solution should be smooth, etc.**

**The statement that the bucket models cannot regularize the LSTM responded to our initial expectations when we were creating the experiments. As we can see in the table added in the previous point, and as stated by the reviewer, the difference in the models can be seen in the head-layer. Our initial expectation was that if our head layer: a) restricts the flexibility of the LSTM because the output of the LSTM (after our dense layer) is further passed through a one-process (single bucket) layer, and b) the one-process layer encodes almost no hydrological process understanding, then the performance of the model would drop. However, this was not the case for the LSTM+Bucket case. In the LSTM+NonSense model, we observed a performance drop of 0.1. However, it is important to note that to be consistent with the other cases, the range in which the LSTM was able to vary the parameters of the NonSense – Conceptual Model was constrained, and the performance drop was caused by hitting those limits. Therefore, our statement that the bucket models cannot regularize the LSTM for the experiments that we propose, answer to the fact that besides allowing extracting untrained variables (if the conceptual model is consistent), the conceptual models are transparent layers that do not change the performance of our model. The LSTM is learning the signal required to predict discharge, and we're not helping (or harming) by adding a local physical operation (the bucket model).**

**Moreover, the fact the models are transparent layers, also indicate that this hybrid approach is unfeasible to select a certain model architecture or a process description over another one, as the LSTM reaches very similar performances with very different even unfeasible head-layers. Therefore, we cannot use this approach in a classical process based manner in which we see different models as hypothesis and select the one with the highest performance as the most realistic one.**

**Specific Comments**

*Line155: "...selected the one that performed the best for each basin. "Was this selection done with train-period, evaluation-period, or test-period data?*

**The selection was done for the test period.**

*Line215: I'm not sure that I understand (or agree with) the distinction being made in this paragraph. The LSTM has a state vector, just like the SHM – there is no difference here. The LSTM requires consecutive predictions in exactly the same way as SHM – again, there is no difference. We use a sequence-to-one training procedure when training the LSTM because this provides more diversity in the mini batch and accelerates training. You could do exactly the same thing for training SHM and/or the hybrid model. You are using a 2-year period for the hybrid model, and you could accomplish the same thing by using a 2-year sequence length for the hybrid model while still training sequence-to-one. You can think about the LSTM as having a 180-day spin up while the hybrid model has a 365 day spin up, and the LSTM is trained seq2one while the hybrid model is trained seq2seq. It's fine if you've found that training the hybrid model using a*

*sequence-to-sequence approach is better, but the way that is motivated (and the way this distinction is framed) in this paragraph is not correct.*

**You are completely right. This paragraph was intended to clarify how we trained the hybrid models, but it is not correct to indicate that the training processes are conceptually different, as both approaches can be trained as seq2one or seq2seq. We will correct this in a revised version of the manuscript.**

*Line230: Do these median statistics come from ensembles (as is used by Kratzert et al),or from single LSTM/Hybrid models?*

**They come from single models.**

*Line235: I think it is too strong to claim that the "consistency and proximity " with Lees "validate the reliability of findings. "This isn't a rigorous way to build on a community benchmark. A better approach is to actually recreate the existing benchmark exactly, which demonstrates that the models are built and trained correctly, then transition the experiment to the basins/ time periods that you want to use for this study. This is, for example, what we did in the Frame et al. papers that required different training/test sets than what were used by previous community benchmarks.*

**To be more rigorous about this we propose the following.**

**In his benchmark, Lees tested his model for 518 basins between 1998-2008. For our case, we tested the model for 60 basins between 2006-2012. Of our 60 basins, 48 are also included in the benchmark, sharing two years of data (2006-2008). Therefore, in a revised version of the manuscript we will compare our results with the ones from the benchmark for the 48 basins and 2 years of data.**

*Line 240: "The LSTM network has the capability to account for biases in the forcing variables (e.g. precipitation or evapotranspiration) because mass conservation is not enforced." We have a paper that demonstrates this explicitly:*

> *Frame, Jonathan M., et al. "On strictly enforced mass conservation constraints for modelling the Rainfall-Runoff process." Hydrological Processes 37.3 (2023): e14847.*

**We will add this reference in a revised version of the manuscript.**

*Line 270: What does the word "overwrite" mean in this sentence? I think that being very specific here about what you want to test is important because how the bucket model influences the LSTM is the central question. There is no such thing as "overwriting". I suggest being crystal clear and precise about what you are envisioning when you ask this question.*

**We agree that overwriting was not a good word choice. As stated in previous responses, our initial idea is that a head layer built without hydrological knowledge would much strongly affect the performance of the hybrid model. Therefore, in this question we wanted to evaluate if the LSTM can operate the head layer such that it overcomes (rather than overwrites) these constraints. We will modify the phrasing in a revised version of the manuscript.**

*Line 300: I agree that the results from section 3.2 don't indicate that a conceptual model can't be used, but again, I want \*some\* type of evidence that this has value. I don't care about subjective arguments or what members of "the community" might think (e.g., line 35), I want some type of real, quantitative, scientific*

*evidence that this is a useful thing to do. Showing strong correlation with soil moisture is interesting, but isn't anything new for these ML-based rainfall-runoff models.*

**With the experiments provided here, besides having access to non-target variables we do not have evidence that suggests other benefits, like an increased global performance or a way to distinguish hydrological meaningful head layers from less meaningful ones. We will explicitly include these statements in a revised version of the manuscript. However, we will conduct future studies in the subject to evaluate if under other conditions the hybrid approaches can provide additional benefits. As discussed in detail in question 1, even though other methods (Lees et al, 2022) have been used to extract non-target information, such approaches require additional training, which makes them different from the methodology proposed here.**

*Line 315: "...suggesting that our model effectively utilizes the well-structured conceptual part to get better predictions of the untrained variables." I disagree — this is one possible explanation for these results, but there is another possibility (that SHM is simply doing nothing), and these experiments can't differentiate between these two possibilities.. These results show that the poorly-structured head layers cause the LSTM to lose information about soil moisture, but there is no indication that the "well-structured" model is being used in any way. The LSTM could be (and I suspect is) providing all of the information about soil moisture here, and the SHM model is doing nothing. A way to test this would be to use the Lees methodology and see what the LSTM can do alone to predict soil moisture. You probably will need to check both the LSTM standalone and the LSTM component of the hybrid model, since the method by which SHM would add value would be through regularization during training, and it is likely (although not guaranteed by the data processing theorem, due to the meteorological inputs to SHM) that the LSTM in the hybrid model will contain all or most of the information about soil moisture that is present in that whole model.*

**Our statement is that the well-structured conceptual model allows us to better predict untrained variables, which we argue is correct, and it is backed up to the fact that the highest correlation of soil moisture is reported for the LSTM+SHM case.**

**As discussed before, the addition of a conceptual model in a head-layer is not improving the performance with respect to the stand-alone LSTM, so in that specific case it can be seen as "the SHM is simply doing nothing". However, the well-structured conceptual model is allowing us to have (without any further training) access to non-target variables with more accuracy than in the other cases, therefore it is doing something. We wanted to extract soil moisture, so we explicitly chose a conceptual model structure that contained a consistent representation of this component, and we were able to reproduce our non-target variable of interest. We are not implying that the soil moisture information is not present in the stand-alone LSTM or even the LSTM part of the conceptual model. We are also not implying that the addition of a well-structured conceptual model boosts the ability of the LSTM to encode soil moisture information coming from the input data. Our only argument is that a well-stablished conceptual architecture allows us to extract untrained variables without any additional training or model. Whether this is meaningful or not is surely open for discussion.**

**We will include this discussion in a revised version of the manuscript.**

*Line 345: The su_max and beta parameters being higher *might* result in lower water availability for down-stream buckets, but really what they are doing is reducing the outflow *rates*. More total water could compensate for this. Is all that is happening here is that the model is pulling a lever to reduce the*

*total output, or is it more complicated - does the rate need to reduce for some reason. What do total inputs (PET) look like over these seasonal cycles? If P-ET is lower for the low-flow seasons, why is it necessary that the flow rates from the unsaturated zone also must be lower? Anyway, it's a little simplistic to just say that higher parameter values result in lower water availability in the downstream buckets.*

**We agree that the effect that the variation of the parameters have in the lower buckets can be compensated by more total water. However, the figure below shows that this was not the case.**

[Figure]

**In this figure we plotted, for basin 10003, the normalized time variation of the beta parameter (green line), and the normalized time variation of the interflow storage (blue line). Here we see a clear pattern that increases in beta are associated with lower water content in the interflow, and that decreases in beta have the opposite effect. Therefore, even though our justification was simplistic, we can see that the parameters are moving water closer to the output in situations where more water is required and reducing the total water content in low flow situations.**

*Line 350: Is there any reason to believe that "noisy" parameter values indicate parameter interaction? Could it just be that there is no low-frequency signal that is needed in this basin to compensate for lack of information in seasonal precipitation signal? Also, what does "noisy" mean? Why do we think this high-frequency variability is noise?*

*This whole set of experiments is leading me to hypothesize that the problem with the bucket models is that there aren't enough buckets. More buckets allow for more flexibility in the mixing of residence times. We could use hundreds of buckets wired in parallel and series, with skip connections (to account for mass transfers between buckets on timescales shorter than the number of timesteps represented by the distance between two buckets in series). We could train this much bigger bucket model and look at residence time mixing ratios over seasons. And really, this would not be very much different than using a set of fully connected layers with linear coefficients, which - surprise, surprise - is exactly what the head layer on the standalone LSTM is.*

**Yes, it can be that the basins that do not present "seasonal patterns" are because no low frequency signal is required. In accordance with what was suggested by the reviewer, in a revised version of the manuscript, we will replace "noisy" with "high-frequency variability".**

**About the second paragraph, we also agree that hundreds of buckets connected in parallel, and series will be similar to an LSTM. Each cell state of the LSTM is modified by a forget gate, input gate and (indirectly) output gate. This is equivalent to having a bucket in which its internal state is updated considering some mass loss (e.g. evapotranspiration), an input (e.g. precipitation) and an outflow (e.g. discharge). The main difference is that the gates in the LSTM depend on the input and the previous states, and in conceptual models, for simplicity, we usually take this as constants. However, this can also**

**by modified by making the gates of the buckets context dependent, and then both models would be equivalent.**

**However, if we have hundreds of buckets, and we want to extract information about non-target variables, similar to the LSTM case, we need an additional model (probe).**

*Line 415: Referring back to my main criticism, I would like to take issue with this sentence from the conclusion section: "This test addressed one of the main benefits of hybrid models over purely data-driven ones, which is their ability to predict untrained variables." I do not believe that it has been shown that this is a benefit of the hybrid modeling approach. LSTMs do this by themselves, and the authors even cited papers in their introduction that demonstrated this. I can imagine thinking that this might be a benefit, but that was not demonstrated in this paper (or any other that is currently published).*

**We argue that the head layer does allow to extract information about untrained variables. The difference between our method and Less et at. 2022 was discussed above.**

*One question I have after reading all this is about why the parameters should change over time (i.e., why dynamic parameters work and static ones don't). Of course, the answer appears to be simply that the bucket model isn't providing any information and since the LSTM is doing the predicting, of course that prediction needs to be dynamic. But I wonder whether the dynamics in the conductivity parameters in particular might vary with moisture content in a some way that could be interpreted as a characteristic curve?*

**In a hypothetical case in which we have a 'perfect' conceptual model, that considers all the processes happening in the basin, the predicted parameters will be constant. However, due to structural limitations of the conceptual architecture, the LSTM does take part in the predicting. Because of how the LSTM and the conceptual model are connected, the only way for the former to pass 'predicting information' to the latter is through the parameters. We argue this is the reason why the parameters change over time. And, yes, it is very well possible that from a detailed analysis of parameter co-variation (with input and with other parameters), a lot could be learned about the underlying hydrological system (but beyond scope here).**

**We will add a related sentence to a revised version of the manuscript.**
* * *
**We thank Grey Nearing for the detailed evaluation of our manuscript. We believe that the changes proposed here will increase the quality of the manuscript and hope we addressed the questions raised in a satisfactory manner.**

---

## Author Response (AR1)

**Author´s response**

Dear Dr. Viviroli,

Please find in attached with this document the review version of our manuscript, which incorporates the changes proposed by the referees.

**Major Changes:**

1. Following the suggestions of the referees, to better validate our ML pipeline, we first compared our data-driven models against existing benchmarks: Lees et al (2021) for CAMELS_GB and Kratzert et al. (2019b) for CAMELS_US. Once this step was completed, we moved forward with the study. We included in the manuscript a new section showing these results.
2. To increase the compatibility of our research with existing studies we modified the training, testing and validation periods to match Lees et al (2022). For this we had to retrain all the models, however it gave us more flexibility to make direct comparisons with their results. As we extended the training period, we were also able to increase the performance of all our models.
3. We included a comparison between the ability of our method to predict non-target variables and the probe method proposed by Lees et al (2022).
4. The codes accompanying the manuscript were modified to increase efficiency and be more user friendly.

**Minor changes:**

5. The abstract was modified.
6. We incorporated the references suggested by both referees.
7. We included the alternative explanation to see hybrid models as head layers, together with a Table explaining the concept.
8. We modified the explanation on how the data-driven models and the hybrid model were trained.
9. We modified the explanation about the time-varying parameters.
10. We presented direct answers to the research questions.

Even with all these changes, the main points of the paper did not change. We believe the modifications made extensively cover the changes proposed by both referees. We would like to thank both, one who remained anonymous and Grey Nearing, as their input in the review process allowed us to produce a better manuscript.

Kind regards,
Eduardo Acuña on behalf of the co-authors.

---

## Author Response (AR2)

**Author´s response**

Dear Dr. Viviroli,

Please find in attached with this document the review version of our manuscript, which incorporates the changes proposed by Grey Nearing.

**Major Changes:**

1. With respect to the comment by referee#2 Grey Nearing:

*I don't agree with the reasoning for performing the experiment on a subset of basins.*

*The reason that the authors state is that they expect conceptual models to not be able to model human-influenced catchments. If there is this (or another) limitation of one type of model in the study, then it seems that this limitation is part of any meaningful comparison. Instead, the approach I would take would be to do the full experiment (on the whole benchmark dataset), and then – if there is a conceptual data split that makes sense – report analyses of results on the full data set and also on that split that you want to highlight.*

*Even if you restrict your analysis to "near-natural" basins, – which, to reiterate, I think is somewhat artificial – you probably should train the LSTM models on the full dataset. 60 catchments is probably not enough for training (see Kratzert et al 2024).*

*Additionally, there is an unfortunate consequence of only using a subset of catchments in that, moving forward, if someone wants to benchmark against or build on your results, they only have a subset of the community benchmark to work with.*

We retrained all our models to consider the whole benchmark dataset (CAMELS-GB). Consequently, the stand-alone LSTM, LSTM+SHM, LSTM+Bucket, LSTM+NonSense, stand-alone SHM, stand-alone Bucket and stand-alone NonSense are now trained in all 669 catchments of CAMELS-GB.

We included section 3.4 in which we tested the models in the original subset of 60 basins, justifying the reason of the subset and its importance there. However, the outcomes reported in the other sections correspond to the results obtained for the whole dataset (669 catchments).

2. We moved explanation about the SHM model and about the training details for the hybrid model to the appendix section. This is intended to make the article easier to read, by moving important but not essential information to the appendix section. The information included in the appendix was already contained in the original manuscript, we only modified the location.

**Minor changes:**

1. With respect to the comment by referee#2 Grey Nearing

*Line 265: It seems like the experiments in this paper could/should be run with ensembles. We do not know whether the conceptual models and hybrid models benefit the same way from ensembles as the pure ML models.*

Using ensembles in hybrid models is indeed an interesting idea. There are multiple ways in which one can do ensembles. For example, different initializations (as done with LSTM) or using multiple conceptual models in parallel and weigh their output. We are currently investigating this effect for

future research; however, we including ensembles is out of the scope for this paper and do not align directly with the main points we want to make.

2. With respect to the comment by referee#2 Grey Nearing

Line 290: Hoge (2022) and Kraft (2022) do not show that fusing deep learning models with hydrological mechanistic models can reach state-of-the-art. The reason for this is that they did not test against current state-of–the-art, and instead tested against handicapped LSTM models that perform significantly worse than current state-of-the-art. As an example, Hoge (2022) used LSTM values from a different study (Jiang, 2022), but they did not use even the best-performing LSTM from that paper, let alone the current SOTA LSTM (which is not from Jiang). Kraftonly tested against physically based models, not ML models, and there is no physically-based hydrology model that is anywhere close to the current SOTA. These papers should not be referenced in the way that they are here.

We modified the references accordingly.

3. With respect to the comment by referee#2 Grey Nearing

Line 293: I am not sure that Mendoza et al discussed dynamic vs. fixed parameters. When they talk about fixed parameters, they mean parameters that are hard-coded in the source code and therefore cannot be calibrated. This is not the same thing as parameters that vary during time series prediction.

We modified the references accordingly.

Even with these changes, the main points of the paper did not change. We believe the modifications made cover the changes proposed by the referee. We would like to thank both referees, one who remained anonymous and Grey Nearing, as their input in the review process allowed us to produce a better manuscript.

Kind regards,
Eduardo Acuña on behalf of the co-authors.